# Skipping the Zeros in Diffusion Models for Sparse Data Generation

Phil Sidney Ostheimer [1]  Mayank Nagda [1]  Andriy Balinskyy [1]  Gabriel Vicente Rodrigues [1]  Jean Radig [2]
Carl Herrmann [2]  Stephan Mandt [3]  Marius Kloft [1]  Sophie Fellenz [1]

## Abstract

Diffusion models (DMs) excel on dense continuous data, but are not designed for sparse continuous data. They do not model exact zeros that represent the deliberate absence of a signal. As a result, they erase sparsity patterns and perform unnecessary computation on mostly zero entries. With Sparsity-Exploiting Diffusion (SED), we model only non-zero values, preserving sparsity. SED delivers computational savings while maintaining or improving generation quality by skipping zeros during training and inference. Across physics and biology benchmarks, SED matches or surpasses conventional DMs and domain-specific baselines, while vision experiments provide intuitive insights into the limitations of dense DMs and the benefits of SED.

## 1. Introduction

Sparsity is a pervasive characteristic of many real-world systems: in the physical world, in living organisms, and in human activity, information manifests sparsely, while most potential states remain unoccupied. As examples, consider particle physics, where only a small fraction of detector cells record energy deposits (Lu et al., 2019; 2021; Howard et al., 2022), recommender systems, where users engage with just a handful of items among millions, or biology, where, e.g. in single-cell RNA (scRNA) sequencing, most measurements are exactly zero and only a limited subset carries signal (Lähnemann et al., 2020). Across such domains, data is not only high-dimensional but also *real-valued and highly sparse*.

The ability to synthesize such data could unlock advances across biology (realistic synthetic cells without expensive scRNA sequencing), physics (fast collision simula-

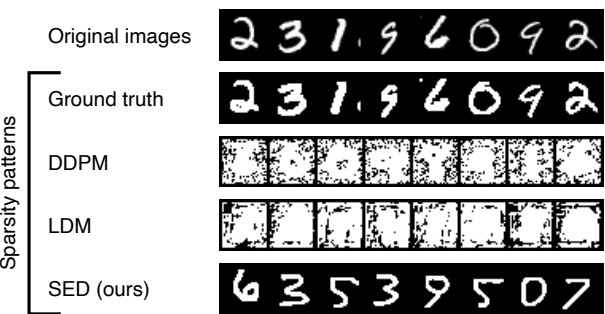

*Figure 1.* Sparsity preservation on MNIST. While dense models (DDPM, LDM) fail to preserve exact zeros and introduce spurious non-zero entries, the proposed Sparsity-Exploiting Diffusion (SED) model preserves sparsity patterns closely aligned with the ground truth.

tion without accelerator runs), and recommender systems (sparsity-aware stress tests), but it requires both *high-fidelity samples* and *faithful recovery of sparsity patterns*. Zeros are semantically meaningful absences of signal (e.g., scRNA dropout events, no energy deposits in particle physics experiments). Failing to preserve them undermines interpretability, trust, and downstream utility (Lähnemann et al., 2020; Qiu, 2020; Lu et al., 2021).

Diffusion models (DMs) (Sohl-Dickstein et al., 2015; Ho et al., 2020; Song & Ermon, 2020; Pandey & Mandt, 2023; Pandey et al., 2025) have emerged as a state-of-the-art for generative modeling, achieving remarkable performance across images, audio, and text. Their stability and ability to capture complex distributions make them a promising candidate for sparse data synthesis. However, existing diffusion approaches are not designed for real-valued sparse data with exact zeros. Standard dense DMs operate over all dimensions and generate continuous values everywhere, failing to preserve exact sparsity patterns and erasing semantically meaningful zeros, as illustrated in Figure 1.

We propose *Sparsity-Exploiting Diffusion (SED)*[1], which exploits sparsity by encoding only non-zero values into a compact latent representation, performing dense diffusion in this space, and then autoregressively reconstructing the

[1]RPTU University Kaiserslautern-Landau [2]Heidelberg University [3]University of California, Irvine. Correspondence to: Phil Sidney Ostheimer <phil.ostheimer@rptu.de>.

*Proceedings of the 43rd International Conference on Machine Learning*, Seoul, South Korea. PMLR 306, 2026. Copyright 2026 by the author(s).

[1]Code is available at https://github.com/PhilSid/sparsity-exploiting-diffusion.

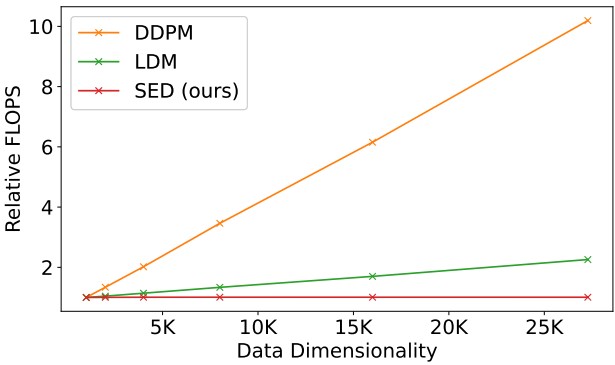

*Figure 2.* SED keeps computational cost nearly constant for generative modeling on high-dimensional sparse scRNA data with a fixed number of active genes. Unlike DDPM and LDM, whose costs grow with total dimensionality, SED processes only non-zero dimensions, maintaining efficiency regardless of input size.

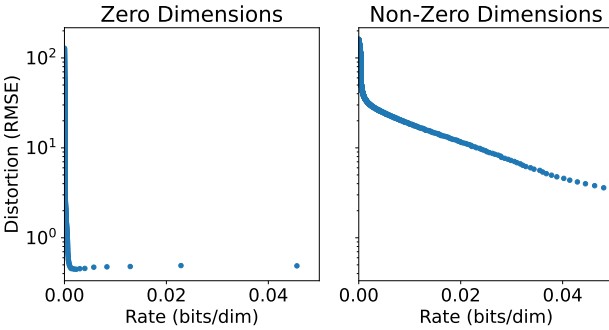

*Figure 3.* DDPM on sparse MNIST images: rate–distortion curves show the allocation of less rate to zero dimensions, yet the denoising network in training/inference processes all dimensions, incurring overhead. The proposed SED operates only on non-zero values, preserving sparsity and avoiding unnecessary compute.

non-zero values. As a result, SED achieves efficiency that scales with the number of non-zeros, in contrast to dense models that scale with the dimensionality (see Figure 2), while preserving sparsity patterns that dense models erase (see Figure 1). Thus, SED enables high-fidelity synthesis of continuous values together with accurate recovery of sparsity patterns.

The key contributions of our paper:

- Sparse-to-dense latent encoding: A latent DM based on a sparsity-aware autoencoder that skips zeros and operates only on non-zero values, avoiding redundant computation in dense DMs.

- Autoregressive sparse decoding: A generation strategy that synthesizes dimension–value pairs only for non-zero entries.

- Extensive empirical validation: In scientific domains where sparsity is fundamental, such as physics and biology, SED achieves quality and efficiency gains over domain-specific and standard DMs.

## 2. Diffusion Models Waste Computation on Zeros

As shown in Figure 1, dense DMs, such as DDPM (Ho et al., 2020) and LDM (Rombach et al., 2022), fail to preserve the sparsity patterns of real-valued data—even on simple datasets like MNIST. This failure is surprising: while these models excel at capturing complex dense distributions, they struggle to maintain exact zeros, a basic structural property of sparse data.

To understand this behavior, we analyze the rate–distortion curve in Figure 3, separately for zero (left) and non-zero (right) dimensions. A rate–distortion curve describes how

reconstruction fidelity improves as more information is allocated: lower distortion indicates higher fidelity, while a higher rate corresponds to greater information capacity devoted to representing the data (Cover & Thomas, 1999; Yang et al., 2023b;c). In DMs, following Ho et al. (2020), the rate at timestep $t$ is defined as the cumulative KL divergence between forward noising distribution $q$ and the model's predicted reverse distribution $p_\theta$ in the ELBO:

$$\sum_{s=t+1}^{T} \mathbb{E}_{q(\mathbf{x}_s|\mathbf{x}_0)}[D_{\mathrm{KL}}(q(\mathbf{x}_{s-1}|\mathbf{x}_s, \mathbf{x}_0) \,\|\, p_\theta(\mathbf{x}_{s-1}|\mathbf{x}_s))].$$

The distortion at timestep $t$ is measured as the root-mean-squared error $\sqrt{\|\mathbf{x}_0 - \hat{\mathbf{x}}_0\|^2 / D}$ between a data point $\mathbf{x}_0 \in \mathbb{R}^D$ and its reconstruction $\hat{\mathbf{x}}_0$.

Figure 3 shows that non-zero dimensions receive substantially more rate, whereas zero dimensions are allocated almost no rate, even at low distortion levels. This behavior aligns with the intuition that dimensions containing little or no information should receive minimal capacity.

However, this allocation reveals a fundamental inefficiency. Despite receiving negligible information capacity, zero dimensions are still fully processed during training and inference: gradient updates and denoising networks operate over the entire input indiscriminately. As a result, dense DMs expend substantial computation on low-rate dimensions while still failing to preserve exact zeros.

Taken together with Figure 1, this analysis highlights a key limitation of dense DMs on sparse data: information capacity is concentrated on informative dimensions, but computation is not. This mismatch motivates our central question—how can DMs avoid unnecessary computation on low-information regions while preserving sparsity? We address this question in Section 4.

## 3. Related Work

We address the problem of efficiently generating high-dimensional sparse continuous data where most values are exactly zero, with the formal problem definition in Section 4. The following paragraphs review existing approaches and highlight their shortcomings when applied to our setting.

**Sparse Data Generation.** Work on sparse data generation divides into continuous and discrete settings. For sparse continuous data, prior deep models target domain-specific applications like calorimeter sensor data, handling sparsity via learnable Dirac delta masses at zero (Lu et al., 2019; 2021), but rely on prior knowledge of, e.g., zero locations using spiral sampling that hinder generalization. For discrete data, Poisson or negative binomial models and deep variants (Zhou & Carin, 2012; Schein et al., 2016; Gong et al., 2017; Guo et al., 2018; Schein et al., 2019; Zhao et al., 2020) capture bursty count sparsity. Yet these probabilistic approaches cannot model our setting of continuous values that are mostly zero. Discrete DMs (Austin et al., 2021; Gu et al., 2022) can generate exact zeros efficiently, but they are also not suited to our setting, as they operate over discrete state spaces and cannot represent real-valued non-zero magnitudes in sparse continuous data. A recent sparsity-aware DM (Ostheimer et al., 2025) adapts standard DMs to sparse data by augmenting each input dimension with a binary sparsity indicator and applying diffusion to the full augmented representation. While this explicitly represents sparsity, it doubles the input size and still performs denoising over all dimensions. SED takes the opposite approach: it removes zero-valued dimensions before diffusion and operates only on dimension-value pairs, so computation scales with the number of non-zero entries rather than the ambient dimensionality.

**Latent Diffusion Models.** Latent diffusion models (LDMs) (Vahdat et al., 2021; Rombach et al., 2022) train DMs in compressed latent spaces via autoencoders, but still process values irrespective of whether they are zero. For 3D shapes, they operate on fixed numbers of points or voxels (Luo & Hu, 2021; Zhao et al., 2021; Zeng et al., 2022) or latent points (Lyu et al., 2023), treating them irrespective of their content, while our method generates only non-zero values, filling the rest with zeros at varying sparsity levels. For language (Lovelace et al., 2023), LDMs autoregressively output variable-length sequences from a fixed-size latent code. However, the output is a sequence with varying length and not a fixed-size vector with varying sparsity levels.

**Transformers.** Transformers treat input data as unordered token sets, requiring explicit positional encodings to capture sequential or spatial relationships: sinusoidal in the original model (Vaswani et al., 2017), learned 2D embeddings in ViTs (Dosovitskiy et al., 2021), and coordinate-based encodings for point clouds (Zhao et al., 2021; Wu et al., 2022; 2024; Draxler et al., 2025). However, all these models process all dimensions of input sets regardless of data content, unlike our approach, which processes only the non-zero values of sparse input vectors. For standard Transformers, the computational and memory cost typically scale quadratically with input length due to the self-attention mechanism. As input dimensions grow, this leads to rapidly increasing resource requirements (Keles et al., 2023).

**Skipping the Zeros.** Zero-skipping in sparse data is a well-established practice, exemplified by sparse matrix representations such as Compressed Sparse Row and Compressed Sparse Column formats that achieve computational efficiency by storing only non-zero values and their corresponding indices. This concept has been adapted for attention-based Transformer models in scRNA representation learning, where redundant computations on predominantly zero-valued data create inefficiencies (Yang et al., 2022). Recent representation learning approaches for scRNA data (Gong et al., 2023; Hao et al., 2024) take inspiration from the masked language modeling objective (Devlin et al., 2019) and employ filtering of masked and zero-valued positions to reduce input size. However, these methods retain index information via gene embeddings and use a Mean Squared Error (MSE) loss only on masked genes to enable representation learning rather than data generation. In particular, positional information is not explicitly encoded as part of a generative process over sparse dimensions, which is essential for maintaining dimensional correspondence and accurately generating sparse real-valued data.

## 4. Sparsity-Exploiting Diffusion (SED)

We now introduce our approach, Sparsity-Exploiting Diffusion (SED). The goal is to efficiently model high-dimensional, continuous data with predominantly exact zeros, where sparsity patterns must be faithfully preserved. To this end, we first formalize the problem and then describe the overall architecture before detailing each component.

**Problem Setup** We model high-dimensional sparse continuous data: Let $\mathcal{X} = \{\mathbf{x}^{(1)}, \ldots, \mathbf{x}^{(n)}\}$ with each $\mathbf{x}^{(i)} \in \mathbb{R}^s$ and $l_i = \|\mathbf{x}^{(i)}\|_0 \ll s$ non-zeros. Unlike discrete sparsity, values are continuous, zeros are exact in $\mathbb{R}$, and the zero pattern must be preserved.

**Overall Architecture** Following the two-stage training paradigm of LDMs (Rombach et al., 2022), SED decomposes sparse data generation into distinct compression and generative phases. Section 4.1 details the first stage: the proposed Sparsity-Aware Variational Autoencoder (SAVAE),

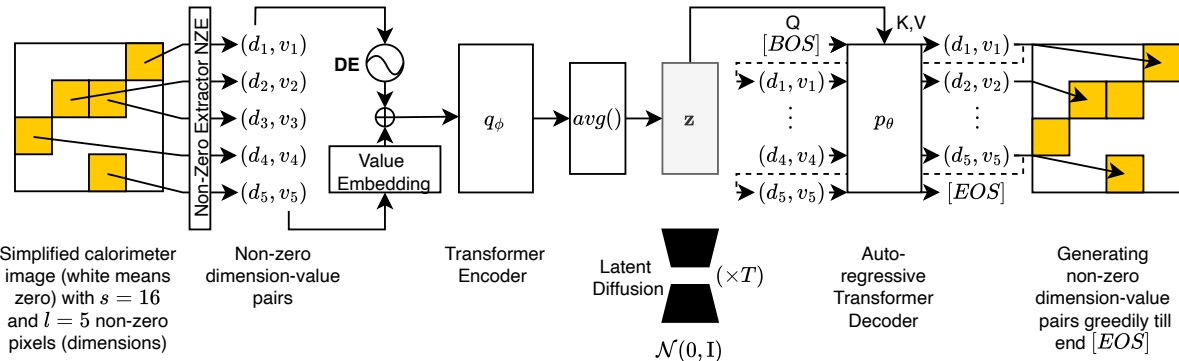

*Figure 4.* The proposed SED processes only non-zero values for efficient sparse data generation. Overview of SED applied to sparse calorimeter images where white pixels represent zero energy deposits. The sparsity-aware encoder $q_\phi$ extracts dimension-value pairs from non-zero input elements, averages the Transformer output to produce a fixed-size dense latent representation $z$, and performs diffusion in this dense space. The autoregressive decoder $p_\theta$ comprises two components: $p_{\theta_1}$ generates multinomial distributions over dimensions and $p_{\theta_2}$ produces Gaussian distributions over values to synthesize dimension-value pairs sequentially.

which employs a Transformer-based encoder that processes only non-zero values to produce a compact, dense latent representation paired with an autoregressive decoder that reconstructs the input. Section 4.2 describes the second stage, integrating SAVAE with latent diffusion for SED. When training the latent diffusion in the second stage, the SAVAE encoder-decoder is kept fixed. The overall architecture and training procedure is illustrated in Figure 4.

### 4.1. Sparsity-Aware Variational Autoencoder

Similar to a standard Variational Autoencoder (VAE) (Kingma & Welling, 2014), we assume a two-step generative process where first a sample $\mathbf{z} \sim \mathcal{N}(0, \mathbf{I})$ is drawn and then a datapoint $\mathbf{x} \sim p_\theta(\mathbf{x}|\mathbf{z})$. However, we do not want to model the distribution over all dimensions of $\mathbf{x}$ as most dimensions have a zero value. We model only those dimensions with non-zero values, resulting in the Sparsity-Aware Variational Autoencoder (SAVAE) and its components:

- **Non-Zero Extractor NZE:** Instead of modeling $\mathbf{x}^{(i)} \in \mathbb{R}^s$ in its full dimensionality, we apply a non-zero extractor NZE to obtain a compact representation $\bar{\mathbf{x}}^{(i)} = \text{NZE}(\mathbf{x}^{(i)}) = (\mathbf{d}^{(i)}, \mathbf{v}^{(i)})$, where $\mathbf{d}^{(i)} = \{j \mid \mathbf{x}_j^{(i)} \neq 0\}$ denotes the indices of dimensions with non-zero values, and $\mathbf{v}^{(i)} = \mathbf{x}_{\mathbf{d}^{(i)}}^{(i)}$ contains the corresponding non-zero values. Both $\mathbf{d}^{(i)}$ and $\mathbf{v}^{(i)}$ are of length $l_i \ll s$, where $l_i = \|\mathbf{x}^{(i)}\|_0$ is the number of non-zero elements in $\mathbf{x}^{(i)}$. This representation differs from conventional Transformer-based encoder-decoder architectures, which typically process all $s$ dimensions of all elements within a sequence regardless of their information content, thereby not exploiting input sparsity for computational efficiency.

- **Dimension Encoding (DE):** There are numerous methods for Transformers to add positional information to understand an element's position (Vaswani et al., 2017; Gehring et al., 2017). In our method, we do not encode the position in a sequence but the dimension index within the input. Therefore, we call it Dimension Encoding (DE). The definition is analogous to positional encodings: $\text{DE}_{(dim, 2i)} = \sin(dim/k^{2i/d_{model}})$ and $\text{DE}_{(dim, 2i+1)} = \cos(dim/k^{2i/d_{model}})$ with $k = 20000$. We add the dimension and the value embedding.

- **Value Embedding:** We employ a linear projection for the value embedding.

- **Encoder $q_\phi$:** SAVAE employs a Transformer-based encoder $q_\phi(\mathbf{z}|\mathbf{d}, \mathbf{v})$ (Vaswani et al., 2017) that approximates the true posterior. Since the number of non-zero entries varies across samples, the encoder produces a variable number of token representations. We therefore apply mean pooling over the Transformer outputs to obtain a single fixed-size representation, which is necessary for the subsequent fixed-dimensional diffusion process. We also evaluated adding a special token to obtain a fixed-size representation, analogous to [CLS] in BERT (Devlin et al., 2019), and observed very similar performance. We use mean pooling for simplicity and training stability. We apply the reparameterization trick and obtain a fixed-size sample $\mathbf{z}$ from the posterior.

- **Decoder $p_\theta$:** At each decoding step $i$, the decoder conditions on the previously generated pairs $(\mathbf{d}_{<i}, \mathbf{v}_{<i})$, which are processed using the same DE and value embedding as in the encoder. The probabilistic decoder $p_\theta(\mathbf{d}, \mathbf{v} \mid \mathbf{z})$ factorizes into two components:

$p_{\theta_1}$, which models a multinomial distribution over the discrete dimension indices $\mathbf{d}$, and $p_{\theta_2}$, which models a Gaussian distribution over the corresponding continuous values $\mathbf{v}$.

**Why autoregressive decoding?**  The autoregressive decoder is a structural requirement of our architectural choice for sparse data modeling. Each sample can contain a different number of non-zero entries, i.e., $l_i = \|\mathbf{x}^{(i)}\|_0$ varies across samples. For example, one scRNA cell may contain only a few active genes, whereas another may contain many more. At generation time, this length is not known in advance: the model must decide both which dimensions are active and when to stop generating dimension–value pairs. We therefore generate pairs sequentially until the decoder emits the end-of-sequence token [EOS]. Importantly, the autoregressive formulation does not require sequential training: we use teacher forcing, so all target dimension-value pairs in a sample can be evaluated in parallel during training. Sequential decoding is only required at sampling time.

**Training Objective.**  We employ a $\beta$-VAE formulation (Higgins et al., 2017) to prevent arbitrarily high-variance latent representations by imposing a KL-divergence regularization with a small $\beta$.

The reconstruction loss is adapted to our zero-skipping representation $\bar{\mathbf{x}} = (\mathbf{d}, \mathbf{v})$ of the input through a composite objective: we minimize the negative log likelihood of observed $\bar{\mathbf{x}} = (\mathbf{d}, \mathbf{v})$ by splitting it into two parts. The first part autoregressively reconstructs $\mathbf{d}$, the second part $\mathbf{v}$. The autoregressive factorization follows a canonical ordering of dimensions, defined as ascending index order, and the decoding process terminates on the end-of-sequence [EOS] token.

This dual-component formulation enables SAVAE to jointly learn both the spatial distribution of informative elements and their associated values, while maintaining a well-structured latent space suitable for subsequent diffusion modeling. The resulting loss is the following:

$$\mathcal{L}_{\text{SAVAE}}(\phi, \theta)$$
$$= \mathbb{E}_{\mathbf{z} \sim q_\phi(\mathbf{z}|\mathbf{d},\mathbf{v})} \left[ -\log p_\theta(\mathbf{d}, \mathbf{v}|\mathbf{z}) \right] \qquad (1)$$
$$+ \beta \cdot D_{\text{KL}}\left( q_\phi(\mathbf{z}|\mathbf{d}, \mathbf{v}) \,\|\, p(\mathbf{z}) \right),$$

where

$$- \log p_\theta(\mathbf{d}, \mathbf{v}|\mathbf{z})$$
$$= - \sum_i \left[ \log p_{\theta_1}(\mathbf{d}_i|\mathbf{d}_{<i}, \mathbf{v}_{<i}, \mathbf{z}) \right. \qquad (2)$$
$$\left. + \log p_{\theta_2}(\mathbf{v}_i|\mathbf{d}_{<i}, \mathbf{v}_{<i}, \mathbf{z}) \right],$$

where $p(\mathbf{z})$ represents the standard normal prior distribution, $D_{\text{KL}}$ denotes the Kullback-Leibler divergence, and

the hyperparameter $\beta$ is set to $\beta = 1 \times 10^{-6}$ following established practice (Rombach et al., 2022) to provide mild regularization without compromising reconstruction quality. The derivation and implementation details are provided in Appendix A and B, respectively. Appendix B.4 further compares SAVAE against a standard VAE, showing that the sparsity-aware representation achieves lower reconstruction error on highly sparse datasets.

### 4.2. Latent Diffusion with Sparsity-Aware Encoding

The trained SAVAE (encoder $q_\phi$ and decoder $p_\theta$) yields a dense latent space that removes structural zeros and redundant spatial detail, letting the likelihood-based DM focus on informative non-zero entries and train in a compact space that scales with signal density rather than input size. The Transformer architecture embeds sparsity-aware inductive biases—using attention over dimension–value pairs and an adapted objective that emphasizes semantically relevant sparse patterns—defined as

$$\mathcal{L}_{\text{SED}}(\theta) = \mathbb{E}\left[ \|\mathbf{z_0} - f_\theta(\mathbf{z_t}, t, \tilde{\mathbf{z}}_0)\|^2 \right], \qquad (3)$$

where $\mathbf{z_0} \sim q_\phi(\mathbf{z}|\mathbf{d}, \mathbf{v})$, $t \sim \mathcal{U}(0, T)$, $\boldsymbol{\epsilon} \sim \mathcal{N}(0, \mathbf{I})$, $\tilde{\mathbf{z}}_0$ is the previous estimate, and $\mathbf{z_t} = \sqrt{\gamma(t)}\mathbf{z_0} + \sqrt{1 - \gamma(t)}\boldsymbol{\epsilon}$ following the standard diffusion forward process with a monotonically decreasing function $\gamma(t)$ from 1 to 0. The neural backbone $f_\theta(\cdot, t, \cdot)$ is implemented as a self-conditioning (Chen et al., 2023) time-conditional U-Net (Ronneberger et al., 2015) with MLP layers rather than convolutional operations, as our dense latent representation $\mathbf{z}$ lacks the grid-like spatial structure that would benefit from convolutions. Since the forward diffusion process is fixed, $\mathbf{z_t}$ can be efficiently computed from $\mathbf{z_0}$ obtained via the encoder $q_\phi$ during training. For generation, samples from $\mathcal{N}(0, \mathbf{I})$ are first denoised and then decoded back to the sparse input space through autoregressive synthesis of dimension-value pairs via decoder $p_\theta$. We employ the log-SNR parameterization (Kingma et al., 2021) for both DDPM (Ho et al., 2020) and DDIM (Song & Ermon, 2020), resulting in SEDP (SED with DDPM sampling) and SEDI (SED with DDIM sampling).

## 5. Experiments

We evaluate SED across six datasets spanning particle physics calorimeter images, scRNA data from biology, and sparse vision tasks, assessing generation quality through domain-specific metrics, computational efficiency, and sparsity preservation.

### 5.1. Experimental Setup and Implementation

**Baselines.**  We compare SED against several *DM baselines*. First, we include DDPM (Ho et al., 2020) and

DDIM (Song & Ermon, 2020), both designed initially for dense data generation. We also evaluate thresholded variants—DDPM-T and DDIM-T—that enforce the training set's average sparsity by zeroing out values smaller than a global threshold in their outputs, providing a straightforward adaptation of dense DMs to sparse domains. As SED is an LDM, we also compare it to non-sparsity-enforcing LDMs (Rombach et al., 2022) and the thresholded variants. Parameter counts are matched to the corresponding LDM baseline within each domain, since both LDM and SED operate in latent space. Details are provided in Appendix B.8. In terms of *domain-specific* baselines, we also compare against SARM D+C (Lu et al., 2021) for particle physics, a sparsity-enforcing model that autoregressively samples data in spiral patterns using prior knowledge of zero locations. For scRNA, we include scDiffusion (Luo et al., 2024), a baseline employing a pretrained autoencoder trained on a large cell corpus without explicit sparsity enforcement. Further architectural and optimization details are in Appendix B, and compute resources are described in Appendix C.

**Datasets.** For particle physics, we utilize calorimeter images from a muon isolation study (Lu et al., 2019), which are characterized by high sparsity (approximately 95%) and are represented as 32×32 pixel grids. Each pixel encodes the energy deposited in a specific calorimeter cell, corresponding to the sum of the transverse momenta ($P_T$) of particles striking that cell. The dataset comprises 33,331 signal images of isolated muons and 30,783 background images where muons are produced in association with jets. In addition, we evaluate our approach on two scRNA datasets: Tabula Muris (Schaum et al., 2018), containing 57K (90% sparse, increasing to 98% when restricted to the 1,000 most highly variable genes), and Human Lung Pulmonary Fibrosis (Habermann et al., 2020), which includes 114K (91% sparse, rising to 96% under the same filtering) cells. For sparse image generation, we employ Fashion-MNIST (Xiao et al., 2017) (50% sparse) and MNIST (LeCun, 1998) (81% sparse), each consisting of 60K grayscale images of size 28×28.

**Evaluation.** Our evaluation assesses sparsity preservation, generation quality using domain-specific metrics, and computational efficiency across three domains. For calorimeter images, we compute the normalized Wasserstein distance $W_P$ between distributions of the sum of momenta transverse to the beam ($P_T$) and invariant mass, comparing the training dataset against 50,000 generated samples following Lu et al. (2019; 2021). For scRNA sequencing data, we employ Spearman Rank Correlation Coefficients (SCC) and Maximum Mean Discrepancy (MMD) (Gretton et al., 2012) to assess the distribution match between real data and 10,000 generated cells, consistent with prior work (Luo et al., 2024). This multi-faceted approach ensures robust

assessment across diverse sparse data modalities while accounting for domain-specific requirements and the unique challenges of evaluating sparsity preservation.

## 5.2. Retaining Sparsity Patterns

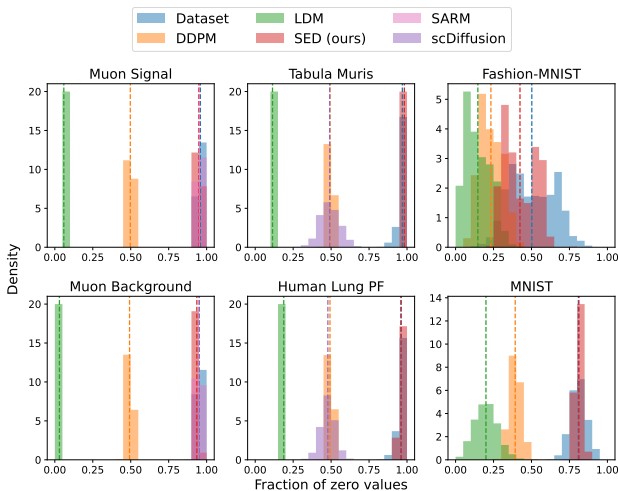

*Figure 5.* Histograms of per-sample sparsity, displaying sparsity levels (20 bins) with mean values indicated by dashed lines. SED achieves accurate sparsity preservation, matching real data sparsity. Sparsity-unaware methods (DDPM, LDM, scDiffusion) systematically underestimate sparsity. On calorimeter images, SED performance is comparable to the sparsity-aware SARM.

Figure 5 evaluates sample-level sparsity preservation. DDPM, LDM, and scDiffusion underestimate sparsity (overly dense samples), with LDMs showing the lowest sparsity levels, not reflecting the dataset's characteristics. In contrast, sparsity-aware SARM and SED closely match ground-truth distributions in both mean and shape. SED also matches the broader distributions and the mean for the sparse image datasets. For scRNA data, the distributions are more peaked, but still reflect per-cell sparsity. Preserving zeros maintains domain structure (e.g., absence of calorimeter detections, scRNA dropouts) and avoids thresholding artifacts in downstream statistics.

## 5.3. High Quality Sparse Data Generation

In the following, we demonstrate SED's capabilities in terms of high-quality sparse data generation, exemplified by sparse calorimeter images. We perform a quantitative and qualitative evaluation.

**Quantitative Evaluation.** Results in Table 1 show standard DMs (DDPM and DDIM) exhibit substantially larger Wasserstein distances $W_p$ for transverse momentum and invariant mass distributions compared to sparsity-aware approaches (SARM and SED). Thresholded variants (DDPM-T and DDIM-T) show improved but inferior performance.

*Table 1.* Quantitative evaluation of sparse calorimeter image generation using Wasserstein distance ($W_P$). Lower values indicate a better distributional match between generated and real data—results for $P_T$ (transverse momentum) and invariant mass distributions for both Muon Signal and Background. SED performs better than input space DMs and SARM, using significantly fewer parameters (15M vs. 37M and 25M) and no domain knowledge. Also, LDMs and their thresholded variants with the same parameter count as SED show substantially larger distributional distances.

| | SIGNAL | | BACKGROUND | |
| MODEL | $\downarrow P_T$ | $\downarrow$ MASS | $\downarrow P_T$ | $\downarrow$ MASS |
|---|---|---|---|---|
| DDPM (37M) | 220.32 | 79.18 | 228.09 | 82.33 |
| DDIM (37M) | 250.89 | 87.30 | 259.30 | 91.22 |
| DDPM-T (37M) | 24.22 | 11.45 | 27.97 | 12.55 |
| DDIM-T (37M) | 25.46 | 12.51 | 31.48 | 13.99 |
| LDMP (15M) | 28.20 | 8.95 | 21.78 | 7.80 |
| LDMI (15M) | 28.21 | 8.95 | 21.77 | 7.80 |
| LDMP-T (15M) | 43.66 | 8.89 | 34.69 | 6.67 |
| LDMI-T (15M) | 43.63 | 8.89 | 34.67 | 6.66 |
| SARM (25M) | 28.01 | 7.49 | 12.61 | 5.66 |
| SEDP (OURS, 15M) | **16.31** | **7.02** | 9.63 | 3.64 |
| SEDI (OURS, 15M) | 17.56 | **7.02** | **9.60** | **3.62** |

LDMs demonstrate higher $W_p$ values, with thresholded variants performing worse except for invariant mass distributions. SED consistently outperforms all methods while using fewer parameters (15M vs. 37M for standard DMs and 25M for SARM) and requiring no domain-specific knowledge (like SARM). SED also achieves better distributional alignment than LDMs with identical parameter count (15M). These results demonstrate that our sparsity-exploiting approach effectively captures sparse patterns in calorimeter data, highlighting the superior architectural fit of SED for sparse data.

**Qualitative Evaluation.** We demonstrate example background images in Figure 6. DDPM-sampled images fail to capture pixel clusters with energy deposits, while thresholded DDPM-T images show isolated single-pixel energy deposits lacking the clustered patterns of real calorimeter data. LDM and the thresholded LDM-T reveal circular energy deposit shapes, suggesting that the model is not able to capture the underlying sparsity patterns but overall round image characteristics. The physics-informed SARM fails to capture energy clusters despite its specialized design. SED successfully generates pixel clusters of deposited energy, demonstrating its ability to model the localized energy patterns characteristic of particle interactions. Signal image samples in Figure 8 (Appendix D) show similar patterns.

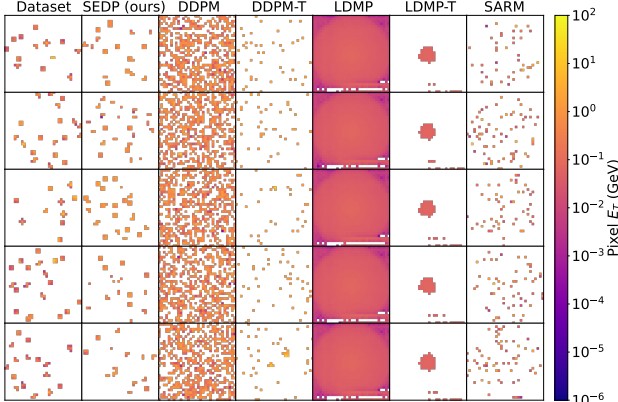

*Figure 6.* SED generates physically realistic sparse background images with proper energy clustering. Comparison of calorimeter images from the Muon Background dataset. Columns show samples from: (1) Dataset, (2) SED (proposed method), (3) DDPM, (4) DDPM with post-hoc thresholding (DDPM-T), (5) LDM, (6) LDM with post-hoc thresholding (LDM-T), and (7) domain-specific SARM. Pixel intensities represent energy deposits in GeV. White indicates zero values. Standard DDPM, LDM, and LDM-T fail to capture localized energy patterns, while DDPM-T and SARM produce unrealistic isolated single-pixel deposits. SED successfully reproduces the clustered energy deposition patterns characteristic of actual particle interactions.

### 5.4. Extremely High Dimensional Sparse Data Generation

We evaluate SED on extremely high-dimensional sparse scRNA datasets containing >95% zero values and up to 27K dimensions. Table 2 presents Tabula Muris and Human Lung Pulmonary Fibrosis results. SED substantially outperforms standard DMs (DDPM, DDIM) and thresholded variants across most metrics, achieving improvements of several orders of magnitude. LDMs demonstrate strong SCC performance but poor MMD scores. Thresholded LDM variants show dramatically degraded SCC performance with only marginal MMD improvements, remaining inferior to SED. Notably, SED surpasses the domain-specific scDiffusion method, which incorporates compute-intensive domain-specific pretraining, on both SCC and MMD metrics. SED's superior scalability to higher dimensions (Section 5.5) makes it particularly well-suited for extremely large and extremely high-dimensional sparse datasets.

### 5.5. Experimental Efficiency Gains over DM Baselines by Skipping Zeros

**Training Compute and Memory for Fixed Non-Zero Dimensions in High-Dimensional Sparse Data.** We analyze training compute and memory for high-dimensional sparse data using scRNA datasets. Using the Human Pulmonary Fibrosis dataset (Habermann et al., 2020), we retain

*Table 2.* Performance comparison on extremely high-dimensional sparse scRNA datasets. SED achieves improvements over standard DMs (DDPM, DDIM) and is superior or on par to thresholded and latent DMs in most settings, while surpassing the domain-specific scDiffusion on both SCC and MMD metrics across all datasets.

| MODEL | TABULA MURIS | | HUMAN LUNG PF | |
|---|---|---|---|---|
| | ↑ SCC | ↓ MMD | ↑ SCC | ↓ MMD |
| DDPM (5M) | 0.50 | 3.60 | 0.30 | 3.34 |
| DDIM (5M) | 0.51 | 3.61 | 0.31 | 3.36 |
| DDPM-T (5M) | 0.55 | **0.34** | 0.31 | 0.70 |
| DDIM-T (5M) | 0.58 | 0.35 | 0.29 | 0.65 |
| LDMP (4M) | **0.87** | 5.82 | **0.86** | 4.94 |
| LDMI (4M) | **0.87** | 5.81 | **0.86** | 4.93 |
| LDMP-T (4M) | 0.26 | 4.24 | 0.31 | 3.76 |
| LDMI-T (4M) | 0.26 | 4.23 | 0.31 | 3.77 |
| SCDIFFUSION (5M) | 0.71 | 1.53 | 0.77 | 1.02 |
| SEDP (OURS, 4M) | 0.74 | 0.55 | 0.82 | **0.54** |
| SEDI (OURS, 4M) | 0.76 | 0.54 | 0.81 | 0.56 |

*Table 3.* The proposed SED demonstrates significant computational efficiency improvements with performance scaling by sparsity level and data structure. Sampling times (milliseconds per sampled batch) show substantial speedups on highly sparse datasets (Muon Signal and Background (Backgr): 95% sparse, MNIST: 81% sparse) compared to less sparse data (Fashion-MNIST (F-MNIST): 50% sparse). scRNA datasets (Tabula Muris (TabulaM), Human Lung Pulmonary Fibrosis (HumanL)) exhibit slight improvements due to architectural differences between implementations.

| DATASET | SPARSITY | SAMPLING | | |
|---|---|---|---|---|
| | | DDPM | LDM | SED |
| SIGNAL | 95% | 453.1 | 94.1 | 23.5 |
| BACKGR | 95% | 452.7 | 93.6 | 24.3 |
| TABULAM | 98% | 9.7 | 8.2 | 7.1 |
| HUMANL | 96% | 9.8 | 8.4 | 7.4 |
| F-MNIST | 50% | 357.7 | 77.7 | 354.6 |
| MNIST | 81% | 358.3 | 77.6 | 52.9 |

1000 highly variable genes and systematically add zero-valued genes to create different dimensionality scenarios. Figures 2 and 9 (Appendix E) show relative compute and memory versus the 1000-gene baseline, Table 11 in Appendix E the absolute baseline runtime. DDPM and LDM exhibit linear scaling with dimensionality, with LDM showing lower constants due to its autoencoder scaling with dimensions, while diffusion operates at fixed dimensionality. SED has almost constant compute and memory behavior while uniquely preserving underlying sparsity patterns (Figure 10, Appendix E), providing distinct advantages for extremely high-dimensional sparse data generation.

**Sampling Runtime.** We analyze sampling runtime performance in Table 3 using a single Nvidia A100, including the 1000 gene baseline for scRNA data. For highly sparse datasets (>80% sparsity), including Muon Signal, Background, and MNIST images, SED demonstrates substantially reduced sampling times compared to DDPM and LDM. However, for the less sparse Fashion-MNIST dataset (50% sparsity), SED exhibits only slight improvements over DDPM, while LDM has the lowest sampling time, reflecting the reduced efficiency gains when sparsity is lower. For scRNA datasets (Tabula Muris and Human Lung Pulmonary Fibrosis) lacking spatial structure, SED shows slightly lower sampling times. This performance difference arises because DDPM and LDM utilize MLP architectures for these tabular datasets, whereas SED's sparsity-optimized approach incurs additional overhead due to the Transformer architecture autoregressively generating the dimension-value pairs.

### 5.6. Visual Inspection and Limitations

**Sparse Image Generation.** Generated images on Fashion-MNIST and MNIST (Figures 11 and 12 in Appendix F) show comparable visual quality across methods. However, SED preserves the underlying sparsity patterns better than standard DM variants (which completely fail) and thresholded variants. Thresholded models (DDIM-T, LDMI-T, DDPM-T, LDMP-T) suppress fine-grained details like edge textures. While thresholding produces visually appealing outputs, it fails to maintain structural properties, particularly suppressing fine-grained details in regions transitioning from all-zero to all-non-zero (and vice versa). These differences highlight that thresholding often fails to preserve key data properties. It also gives insights into why thresholding does not always produce the best results in our experiments on scientific data. SED's sparsity-aware architecture more reliably captures visual fidelity and distributional properties by focusing computational resources on informative non-zero dimensions.

**Limitations.** SED is trained exclusively on correctly ordered dimension orderings and, in the vast majority of cases, also generates correct orderings during sampling. Figure 7 illustrates error cases where it generates incorrect dimension orderings during sampling. Table 4 quantifies the prevalence of correctly ordered samples across datasets. For most applications, SED produces valid dimension orderings in more than 98.5% of samples. Particle physics datasets are more challenging: SED achieves correct orderings in 95.5% of signal and 87.9% of background samples, likely due to the highly structured combination of clustered and isolated energy deposits. We do not observe systematic degradation with longer sequences. Table 4 provides evidence against this concern: Fashion-MNIST, which has the longest sequences among our benchmarks due to its lower sparsity,

achieves 100% correct dimension orderings. In contrast, the highest error rates occur on the muon datasets, which have much shorter non-zero sequences. This suggests that decoding difficulty depends more on the complexity of the data distribution than on sequence length alone. Importantly, as shown in Section 5.3, these error cases do not compromise SED's ability to generate high-quality calorimeter images, as those samples were not filtered out.

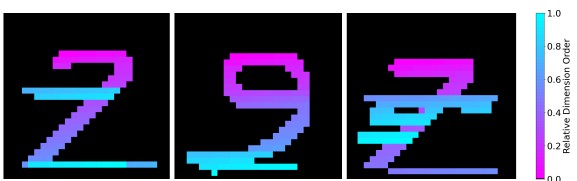

*Figure 7.* SED's greedy autoregressive dimension generation produces correct orderings in the vast majority of cases. Rare failures occur when dimensions are generated out of order, which can lead to unrealistic samples, as illustrated here on MNIST.

*Table 4.* SED's prevalence of valid (ascending) dimension orderings across Muon Signal (S), Muon Background (BG), Tabula Muris (TM), Human Lung Pulmonary Fibrosis (HL), Fashion-MNIST (FM), and MNIST (MN). Correct ordering rates are high across all domains, with lower rates on particle physics datasets.

| MODEL | S | BG | TM | HL | FM | MN |
|---|---|---|---|---|---|---|
| SEDP | 95.6% | 87.9% | 98.7% | 99.7% | 100.0% | 99.3% |
| SEDI | 95.5% | 87.9% | 98.5% | 99.6% | 100.0% | 99.3% |

## 6. Conclusion and Future Work

SED enables diffusion-based generation of high-dimensional sparse continuous data while preserving exact sparsity by diffusing only representations of non-zero dimension-value pairs. Its computational benefits are most pronounced in the high-dimensional, highly sparse regime targeted by this work, where dense DMs spend substantial computation on dimensions that are exactly zero. In most of the considered benchmarks, SED matches or surpasses dense DMs and domain-specific baselines using fewer or comparable parameters and no handcrafted priors or costly pre-training. For lower-sparsity data, such as Fashion-MNIST, these efficiency gains naturally diminish because more non-zero entries must be modeled explicitly.

At the same time, SED highlights key challenges in generative modeling for sparse data. Its variable-length dimension-value representation requires autoregressive decoding and an explicit ordering of dimensions, which can introduce sequential sampling overhead and occasional ordering errors. These limitations are particularly visible on scRNA data, where the baselines show superior performance. Future

work should therefore explore complementary mechanisms that preserve exact zeros while reducing reliance on autoregressive decoding, fixed orderings, or sequential dimension generation.

More broadly, this work reflects a modeling perspective that is common in many scientific domains but less emphasized in modern generative modeling, where benchmarks and architectures are often implicitly dense. Our results align with the *sparsity-of-effects principle* (Box et al., 1978; Donoho, 2006): many complex systems contain only a small subset of active factors, while the rest are absent. Future work should further investigate generative models that explicitly account for both presence and absence, improving scalability, robustness, and sampling efficiency for sparse scientific data.

## Impact Statement

This work contributes to the development of more computationally efficient and structurally faithful generative models for sparse scientific data, with potential positive impact in domains such as particle physics and biology where sparsity is inherent and meaningful. By encouraging generative models that explicitly distinguish between presence and absence, this work may help shift common modeling assumptions in machine learning toward closer alignment with scientific data characteristics, supporting more reliable simulation and analysis workflows. We do not anticipate immediate ethical risks specific to this method. However, as with all generative approaches, synthetic data should be used carefully to avoid misinterpretation in high-stakes scientific settings. This work does not directly address fairness or bias considerations, which remain important directions for future research as sparse generative models are applied more broadly.

## Acknowledgments

This work was conducted within the initiative AI-Care by the Carl-Zeiss Foundation (PSO, AB, JR, CH, MK, SF). MK and SF further acknowledge support by the DFG through FOR 5359 (ID 459419731), TRR 375 (ID 511263698), SPP 2298 (IDs 441826958 and 441826958), and SPP 2331 (IDs 441958259, 553345933, and 466468799), as well as by the BMFTR award 01IS24071A. SM further acknowledges funding from the National Science Foundation (NSF) through an NSF CAREER Award IIS-2047418, IIS2007719, the NSF LEAP Center, and the Hasso Plattner Research Center at UCI. The simulations/calculations were partly executed on the high performance cluster "Elwetritsch" at the RPTU Kaiserslautern-Landau which is part of the "Alliance of High Performance Computing Rheinland-Pfalz" (AHRP). We kindly acknowledge the support of RHRZ.

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

## A. SAVAE Objective

In this section, we detail the realization of our SAVAE loss in (1), and especially its reconstruction term:

$$
\begin{aligned}
& -\log p_\theta(\mathbf{d}, \mathbf{v}|\mathbf{z}) \\
= {} & -\sum_i \log p_{\theta_1}(\mathbf{d}_i|\mathbf{d}_{<i}, \mathbf{v}_{<i}, \mathbf{z}) \\
& -\sum_i \log p_{\theta_2}(\mathbf{v}_i|\mathbf{d}_{<i}, \mathbf{v}_{<i}, \mathbf{z})
\end{aligned}
\tag{4}
$$

The first part of the equation for the discrete tokens just becomes a negative log likelihood loss of a multinomial logistic regression at step $i$. For the second part, we model $p_{\theta_2}(\mathbf{v}_i|\mathbf{d}_{<i}, \mathbf{v}_{<i}, \mathbf{z})$ as $\mathcal{N}(\mathbf{v}_i; \boldsymbol{\mu}_i, \boldsymbol{\Sigma}_i)$ and get

$$
\begin{aligned}
& -\sum_i \log p_{\theta_2}(\mathbf{v}_i|\mathbf{d}_{<i}, \mathbf{v}_{<i}, \mathbf{z}) \\
= {} & -\sum_i \log \mathcal{N}(\mathbf{v_i}; \boldsymbol{\mu}_i, \boldsymbol{\Sigma}_i) \\
= {} & -\sum_i \left[ \frac{1}{2}(\mathbf{v}_i - \boldsymbol{\mu}_i)^\top \boldsymbol{\Sigma}_i^{-1}(\mathbf{v}_i - \boldsymbol{\mu}_i) + \frac{1}{2}\log\left|(2\pi)\,\boldsymbol{\Sigma}_i\right| \right]
\end{aligned}
\tag{5}
$$

In practice, we further simplify this by treating each $\boldsymbol{\Sigma}_i$ as isotropic with fixed variance (i.e., $\boldsymbol{\Sigma}_i = \sigma^2 I$) and discarding the constant log-determinant term, so that up to an additive constant the negative log-likelihood becomes proportional to

$$
\sum_i \|\mathbf{v}_i - \boldsymbol{\mu}_i\|^2,
\tag{6}
$$

i.e. the familiar MSE. This MSE simplification makes sense because, under a homoscedastic Gaussian assumption with fixed variance, the only term that depends on the model parameters is the squared deviation, and dropping constants both reduces computational overhead and yields a convex surrogate that is straightforward to optimize.

## B. Implementation Details and Hyperparameters

The code for all models and experiments is available at https://github.com/PhilSid/sparsity-exploiting-diffusion.

### B.1. Architecture

- **SAVAE** We use a standard Transformer architecture (Vaswani et al., 2017) for the SAVAE encoder-decoder. During training, we employ teacher forcing and greedy decoding for sampling. We detail the hyperparameters in Appendix B.3.

- **SED U-Net** As SED's latent representations do not exhibit the same spatial structure as input space DMs (Ho et al., 2020; Song & Ermon, 2020) or LDMs (Rombach et al., 2022; Vahdat et al., 2021), we cannot apply Convolutional Neural Networks (CNNs) but resort to Multi-Layer Perceptrons (MLPs). We detail the hyperparameters in Appendix B.5.

- **Baselines**
  - **DMs** For calorimeter and sparse image generation tasks, we employ CNN-based U-Nets following established protocols (Ho et al., 2020; Ronneberger et al., 2015). For scRNA data generation, we adopt a Multi-Layer Perceptron (MLP) with skip connections, following scDiffusion (Luo et al., 2024), as scRNA data lacks a spatial structure, making convolutional operations inappropriate. Implementation details can be found in Appendix B.6.
  - **LDMs** For calorimeter and sparse image generation tasks, we employ CNN-based VAEs and U-Nets following established protocols (Rombach et al., 2022; Vahdat et al., 2021). For scRNA data generation, we adopt a Multi-Layer Perceptron (MLP) with skip connections, following scDiffusion (Luo et al., 2024) for the VAE as well as the U-Net, as scRNA data lacks a spatial structure, making convolutional operations inappropriate. Implementation details can be found in Appendix B.7.

## B.2. Optimization and Sampling

We train all DMs for 500K steps using Adam (Kingma & Ba, 2015) for optimization with a constant learning rate of 0.0001. Following the standard LDM training procedure (Vahdat et al., 2021; Rombach et al., 2022; Yang et al., 2023a), we train SAVAE and the latent denoising network in two stages. The SAVAE parameters are frozen when training the diffusion model. This avoids optimizing the reconstruction and diffusion losses jointly, which would introduce an additional loss-balancing problem. We consider end-to-end training an interesting direction for future work. For SAVAE, we train for 100K steps and use a linearly increasing and afterwards exponentially decreasing learning rate (standard for Transformers (Vaswani et al., 2017)). An exponential moving average of the parameters is used to improve training dynamics with a decay factor of 0.9999. We use a fixed number of 1000 steps for DDIM and DDPM sampling and greedy sampling to determine the next dimension-value pair.

## B.3. SAVAE Hyperparameters

We summarize SAVAE hyperparameter settings in Table 5. For the Transformer-based architecture, we use the same terminology as Vaswani et al. (2017). All models are trained for up to 100,000 steps with early stopping. We use the same model size for the two particle physics datasets, Muon Signal, Muon Background, and for the two sparse vision datasets Fashion-MNIST, and MNIST—which have similarly shaped inputs. We opt for a smaller model variant for the biology task on scRNA datasets Tabula Muris and Human Lung Pulmonary Fibrosis, as these inputs have comparable dimensions but are markedly sparser.

*Table 5.* Hyperparameter settings for SAVAE.

| HYPERPARAMETER | PHYSICS/VISION | BIOLOGY |
|---|:---:|:---:|
| $d_{\text{MODEL}}$ | 256 | |
| $d_{\text{FF}}$ | 1024 | |
| NUM. HEADS | 4 | |
| BATCH SIZE | 128 | |
| DROPOUT | 0.1 | |
| BETA | 1E-06 | |
| ITERATIONS | 100K | |
| NUM. LAYERS | 6 | 3 |
| $N_{\text{PARAMS}}$ | 12M | 6M |

## B.4. SAVAE vs. Standard VAE Reconstruction

To assess whether the sparse representation used by SAVAE sacrifices reconstruction fidelity, we compare its reconstruction MSE against a standard VAE trained on the same dataset. Table 6 reports both training and validation MSE. SAVAE achieves lower reconstruction error on the scientific highly sparse datasets and remains competitive on sparse image data, with Fashion-MNIST illustrating that the benefit diminishes when sparsity is lower.

*Table 6.* Reconstruction MSE comparison between a standard VAE and the proposed SAVAE. SAVAE achieves lower reconstruction error on highly sparse datasets, while Fashion-MNIST is the main exception due to its lower sparsity and longer effective output sequences.

| | | TRAIN MSE | | VALIDATION MSE | |
|---|:---:|:---:|:---:|:---:|:---:|
| DATASET | SPARSITY | VAE | SAVAE | VAE | SAVAE |
| SIGNAL | 95% | 3.3E-6 | **2.5E-6** | 3.5E-5 | **3.3E-5** |
| BACKGROUND | 95% | 4.5E-6 | **4.1E-6** | 7.2E-6 | **6.5E-6** |
| TABULA MURIS | 98% | 1.8E-4 | **0.4E-4** | 1.9E-4 | **0.4E-4** |
| HUMAN LUNG PULMONARY FIBROSIS | 96% | 1.2E-4 | **0.6E-4** | 1.3E-4 | **0.6E-4** |
| FASHION-MNIST | 50% | **7.0E-3** | 8.1E-3 | **7.3E-3** | 8.4E-3 |
| MNIST | 81% | 5.1E-3 | **3.4E-3** | **3.2E-3** | 3.75E-3 |

## B.5. SED Hyperaparameters

We summarize SED hyperparameter settings in Table 7. For the sparse vision images of MNIST and Fashion-MNIST as well as for the Muon Signal and Background calorimeter images, we use a model to match the baseline's number of parameters (37M). For scRNA datasets, we use a model to match the baselines' number of parameters for DDPM/DDIM.

*Table 7.* Hyperparameter settings for MLP-based SED U-Nets for denoising.

| HYPERPARAMETER | PHYSICS/VISION | BIOLOGY |
|---|---|---|
| $|\mathcal{Z}|$ | 1000 | |
| DIFF. STEPS | 1000 | |
| NOISE SCHEDULE | COSINE | |
| ARCHITECTURE | MLP | |
| HID. DIM. | 1024,768, 512,512, 256,256, 128 | 512,512, 256,128 |
| DROPOUT | 0.1 | |
| BATCH SIZE | 128 | |
| ITERATIONS | 500K | |
| $N_{\text{PARAMS}}$ | 15M | 4M |

## B.6. DM Baselines

We apply the same U-Net architecture across all baseline models to ensure fair comparison. Given the similar spatial dimensions and structural properties of both calorimeter images and sparse images, they share the same CNN-based architecture and use MLPs for scRNA data. Following prior work (Ho et al., 2020; Luo et al., 2024), we summarize the hyperparameter settings in Table 8.

*Table 8.* Hyperparameter settings for the baselines DDPM, DDIM, DDPM-T, and DDIM-T.

| HYPERPARAMETER | PHYSICS/VISION | BIOLOGY |
|---|---|---|
| $\mathbf{x}$-SHAPE | 32×32/28×28 | 1000 |
| DIFFUSION STEPS | 1000 | |
| NOISE SCHEDULE | COSINE | |
| ARCHITECTURE | CNN | MLP |
| BASE CHANNEL SIZE | 256 | 512 |
| CH. MULT./HID. DIM. | 1,1,1 | 512,512,256,128 |
| DROPOUT | 0.1 | |
| BATCH SIZE | 256 | |
| ITERATIONS | 500K | |
| $N_{\text{PARAMS}}$ | 37M | 5M |

## B.7. LDM Baselines

We apply the same VAE and U-Net architecture across all LDM baseline models to ensure fair comparison. Given the similar spatial dimensions and structural properties of both calorimeter images and sparse images, they share the same CNN-based architecture and use MLPs for scRNA data. Following prior work (Ho et al., 2020; Luo et al., 2024), we summarize the hyperparameter settings in Table 9 for the deployed VAEs and in Table 10 for the deployed U-Nets.

## B.8. Parameter Count Matching

We match parameter counts to the corresponding latent diffusion baseline within each domain, which provides the most controlled comparison because both SED and LDM are latent diffusion methods. For physics and vision datasets, SED and

*Table 9.* VAE hyperparameter settings for LDMP, LDMI, LDMP-T, and LDMI-T.

| HYPERPARAMETER | PHYSICS/VISION | BIOLOGY |
|---|---|---|
| **x**-SHAPE | 32×32/28×28 | 1000 |
| **z**-SHAPE | 8×8/7×7 | 64 |
| ARCHITECTURE | CNN | MLP |
| BASE CHANNEL SIZE | 192 | 128 |
| CH. MULT./HID. DIM. | 2,4 | 512,512,256,128 |
| BATCH SIZE | 256 | |
| DROPOUT | 0.1 | |
| BETA | 1E-06 | |
| ITERATIONS | 100K | |
| $N_{\text{PARAMS}}$ | 12M | 6M |

*Table 10.* U-Net hyperparameter settings for LDMP, LDMI, LDMP-T, and LDMI-T.

| HYPERPARAMETER | PHYSICS/VISION | BIOLOGY |
|---|---|---|
| **z**-SHAPE | 8×8/7×7 | 64 |
| $|\mathcal{Z}|$ | 64/49 | 64 |
| DIFFUSION STEPS | 1000 | |
| NOISE SCHEDULE | COSINE | |
| ARCHITECTURE | CNN | MLP |
| BASE CHANNEL SIZE | 192 | 128 |
| CH. MULT./HID. DIM. | 1,1 | 512,256,128 |
| DROPOUT | 0.1 | |
| BATCH SIZE | 256 | |
| ITERATIONS | 500K | |
| $N_{\text{PARAMS}}$ | 15M | 4M |

LDM use the same size with 15M parameters. The input-space DM baseline uses 37M parameters because full-resolution CNNs require higher capacity, while the domain-specific SARM baseline uses 25M parameters. For biology datasets, SED and LDM both use 4M parameters, while the input-space DDPM baseline uses 5M parameters.

Under this controlled comparison, SED consistently outperforms LDM at equal parameter count across the reported datasets and metrics. Exact parameter matching across fundamentally different architectures, such as CNNs, MLPs, and Transformers, is not generally meaningful because the same number of parameters can correspond to different effective model capacities and inductive biases.

## C. Compute Resources

We executed the experiments on two kinds of servers:

- NVIDIA DGX-A100 server

  - 8×A100 (40GB)
  - AMD Epyc 7742 CPU with 64 cores
  - 2TB main memory
  - NVIDIA DGX Server Version 7.1.0 (GNU/Linux 6.8.0-60-generic x86_64)
  - torch=2.7.1, lightning=2.5.1post0

- NVIDIA DGX-1 server

  - 8×V100 (32GB)
  - 40 cores

- 512GB main memory
- Red Hat Linux with 4.18.0-553.62.1.el8_10.x86_64
- torch=2.7.1, lightning=2.5.1.post0
- SLURM-managed execution with maximum 6 CPU cores, 100GB memory reserved

All baselines, SED training, and sampling benchmarking (runtime) for all datasets were executed on the A100. For quantitative performance comparisons, we executed the particle physics and sparse image generation experiments on the A100 server, while the smaller scRNA SED experiments ran on the DGX-1 server. Due to the deployed diffusion models' large scale and high computational cost, each experiment was performed only once. The substantial GPU resources and time required—especially for sampling—made multiple runs or significance testing infeasible. Although variance analysis is important, practical constraints limited our evaluation to a single run per experiment.

## D. High Quality Sparse Data Generation Detailed Results

In addition to the sample calorimeter images for the Muon Background dataset in Section 5.3, we also present the results for the Muon Signal dataset in Figure 8. The results mostly reflect those for the signals: DDPM-sampled background images fail to capture coherent clusters of energy deposits, producing dispersed and unrealistic patterns. The thresholded DDPM-T images display numerous isolated single-pixel energy deposits lacking the spatial coherence observed in real calorimeter data. LDM and the thresholded LDM-T reveal an indistinguishable noise across the image, suggesting that the model is not able to capture the underlying sparsity patterns. Our proposed SED effectively reproduces the characteristic clustered energy patterns in background images, demonstrating its ability to model the subtle and complex localized energy distributions typical of these events. The domain-knowledge-intensive SARM model again fails to generate coherent energy clusters despite its specialized physics-informed design.

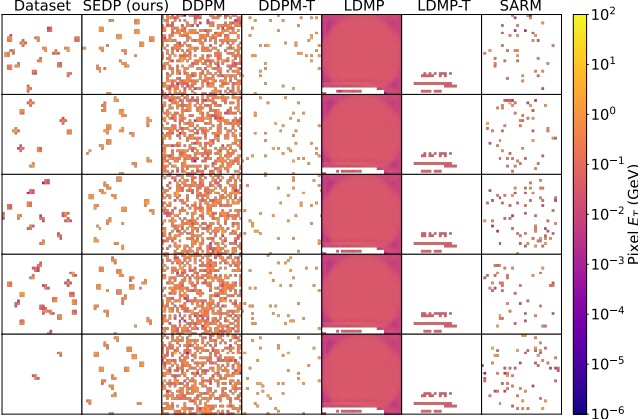

*Figure 8.* SED generates physically realistic sparse calorimeter images with proper energy clustering. Comparison of generated signal calorimeter images from muon isolation dataset. Columns show: (1) real training data, (2) SED (proposed method), (3) DDPM, (4) DDPM with post-hoc thresholding (DDPM-T), (5) LDM, (6) LDM with post-hoc thresholding (LDM-T), and (7) SARM. Pixel intensities represent energy deposits in GeV; white indicates zero values. Standard DDPM, LDM, and LDM-T fail to capture localized energy patterns, while DDPM-T and SARM produce unrealistic isolated single-pixel deposits. SED successfully reproduces the clustered energy deposition patterns characteristic of actual particle interactions.

## E. Experimental Efficiency Gains Detailed Results

In this section, we present additional results for Section 5.5 on the computational efficiency of SED in settings with extremely high-dimensional data and a high number of zero dimensions and the resulting sparsity levels of generated data. Figure 9 shows relative memory versus the 1000-gene baseline. DDPM and LDM exhibit linear memory scaling with dimensionality, with LDM showing lower constants due to its autoencoder scaling with dimensions while diffusion operates at fixed dimensionality. SED demonstrates almost constant memory behavior.

Figure 10 illustrates the sparsity patterns produced by the models compared to the pre-processed dataset with varying sparsity levels. SED closely mirrors the underlying sparsity distributions, even in settings where the sparsity exceeds 99%,

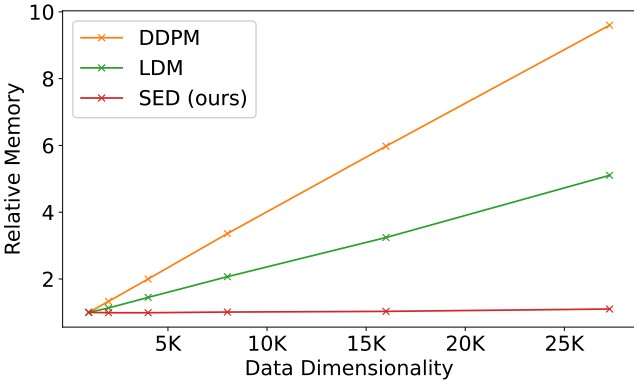

*Figure 9.* SED keeps memory usage nearly constant for high-dimensional (1K–27K dimensions) sparse data with a fixed number of active genes (1K). Unlike DDPM and LDM, whose costs grow with total dimensionality, SED processes only the non-zero gene expression values, maintaining efficiency regardless of input size.

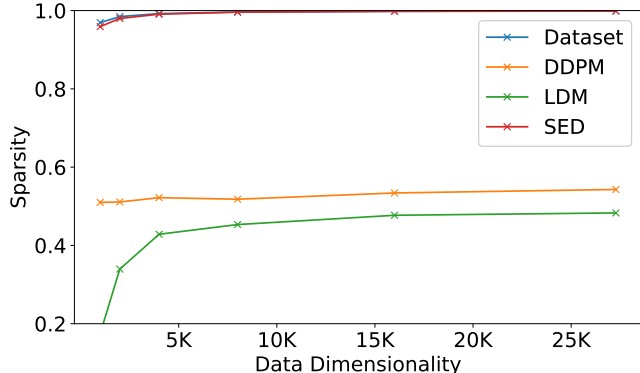

*Figure 10.* Sparsity levels for Human Lung Pulmonary Fibrosis with varying ground-truth sparsity (1K–27K dimensions; 1K active genes fixed). The plot compares DDPM, LDM, and SED, showing that SED is the only model to accurately reflect sparsity even beyond 99%.

effectively capturing zero-valued elements' proportion and structure. In contrast, DDPM and LDM fail to reproduce these sparsity patterns, resulting in a distribution divergent around 50 %. We also excluded DDPM-T from this comparison since it is a post-hoc thresholded variant of DDPM and therefore exhibits identical runtime behavior. These results underscore the advantage of our sparsity-aware SED in preserving critical structural properties of highly sparse scRNA data.

We analyze training runtime performance in Table 11 using a single Nvidia A100. For highly sparse datasets (>80% sparsity), including Muon Signal, Background, Tabula Muris and Human Lung Pulmonary Fibrosis, and MNIST images, SED demonstrates substantially reduced training times compared to DDPM and LDM. However, for the less sparse Fashion-MNIST dataset (50% sparsity), SED exhibits higher training time, reflecting the reduced efficiency gains when sparsity is lower.

## F. More Insights from Spare Image Generation

Generated images on Fashion-MNIST and MNIST (Figures 11 and 12) appear visually similar across models, but closer inspection highlights important differences in sparsity preservation and structural detail. SED outperforms both standard and thresholded approaches by more faithfully reflecting the datasets' true sparsity patterns and maintaining critical details at sparse transitions. Thresholded models (DDIM-T, DDPM-T) tend to produce smoother, visually clean images, but this comes at the expense of finer edge textures in Fashion-MNIST and nuanced stroke width variations in MNIST, as thresholding only imposes sparsity as a post-hoc step rather than during generation. In contrast, SED's sparsity-aware architecture explicitly models and reconstructs only non-zero values and their positions, resulting in better preservation of connected structures, edge details, and the true distribution of sparsity—especially within challenging regions where structural zeros and informative pixels are intermixed. This principled approach enables SED to generate sparse images that

*Table 11.* The proposed SED demonstrates significant computational efficiency improvements with performance scaling by sparsity level. Training times (milliseconds per training batch) show substantial speedups on highly sparse datasets (Muon Signal, Muon Background, Tabula Muris, Human Lung Pulmonary Fibrosis: >95% sparse, MNIST: 81% sparse) compared to moderately sparse data (Fashion-MNIST: 50% sparse).

| | | TRAINING | | |
|---|---|---|---|---|
| DATASET | SPARSITY | DDPM | LDM | SED |
| SIGNAL | 95% | 150.3 | 130.4 | 40.1 |
| BACKGROUND | 95% | 151.0 | 131.3 | 40.1 |
| TABULA MURIS | 98% | 10.2 | 10.1 | 3.0 |
| HUMAN LUNG PULMONARY FIBROSIS | 96% | 10.4 | 10.3 | 3.0 |
| FASHION-MNIST | 50% | 123.8 | 161.5 | 280.1 |
| MNIST | 81% | 120.9 | 160.4 | 101.8 |

not only look realistic but also retain the nuanced patterns crucial for downstream tasks.

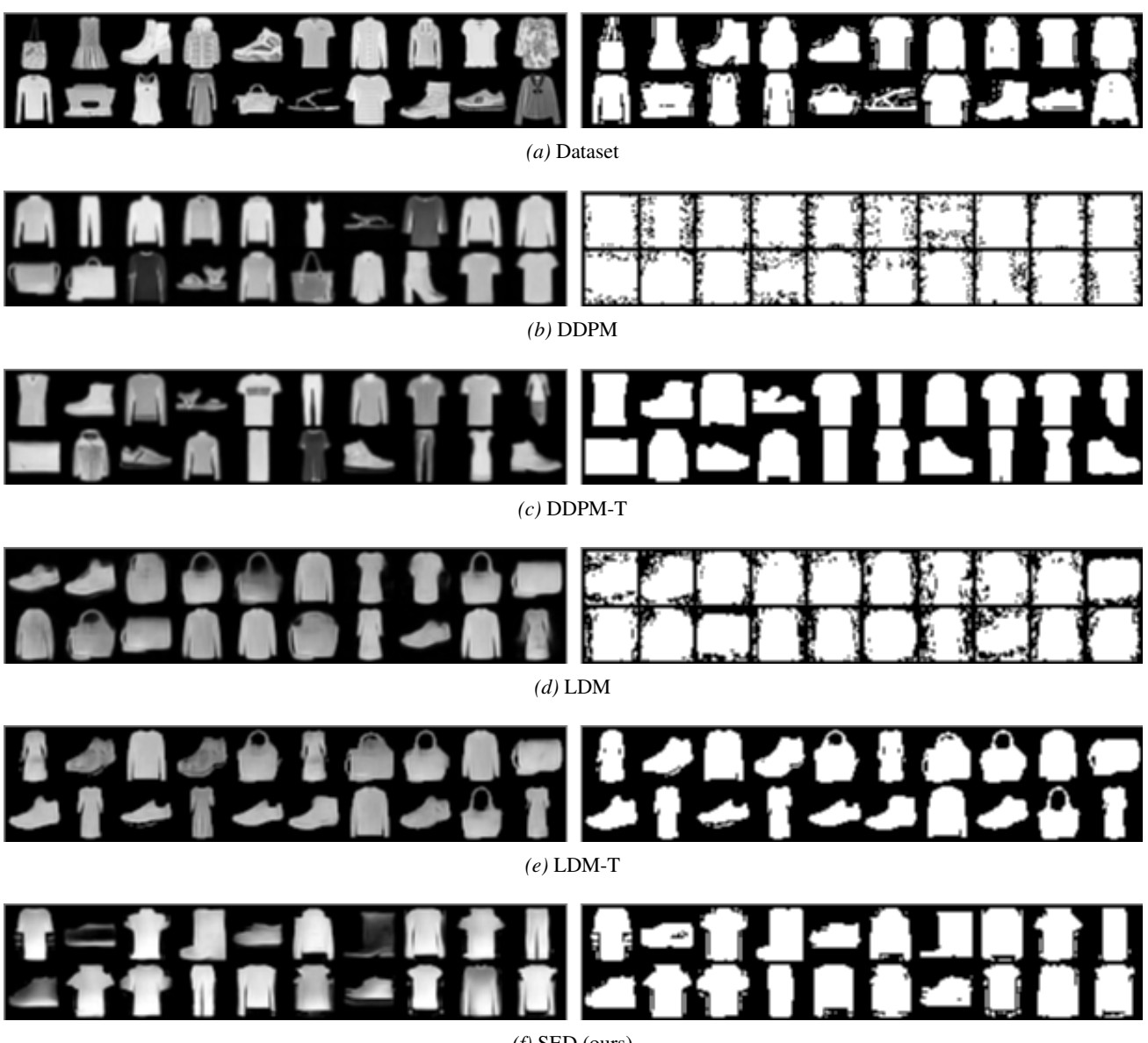

*(a)* Dataset

*(b)* DDPM

*(c)* DDPM-T

*(d)* LDM

*(e)* LDM-T

*(f)* SED (ours)

*Figure 11.* Shown are, from top to bottom, in the first row: Fashion-MNIST images sampled from the dataset, DDPM sampled images, thresholded DDPM sampled images (DDPM-T), LDM sampled images, thresholded LDM sampled images (LDM-T), and SED sampled images. The second columns contains the respective sparsity information. Despite highly visually similar images, DDPM and LDM fail to reflect the sparsity, whereas the proposed SED has a similar sparsity to the images from the dataset.

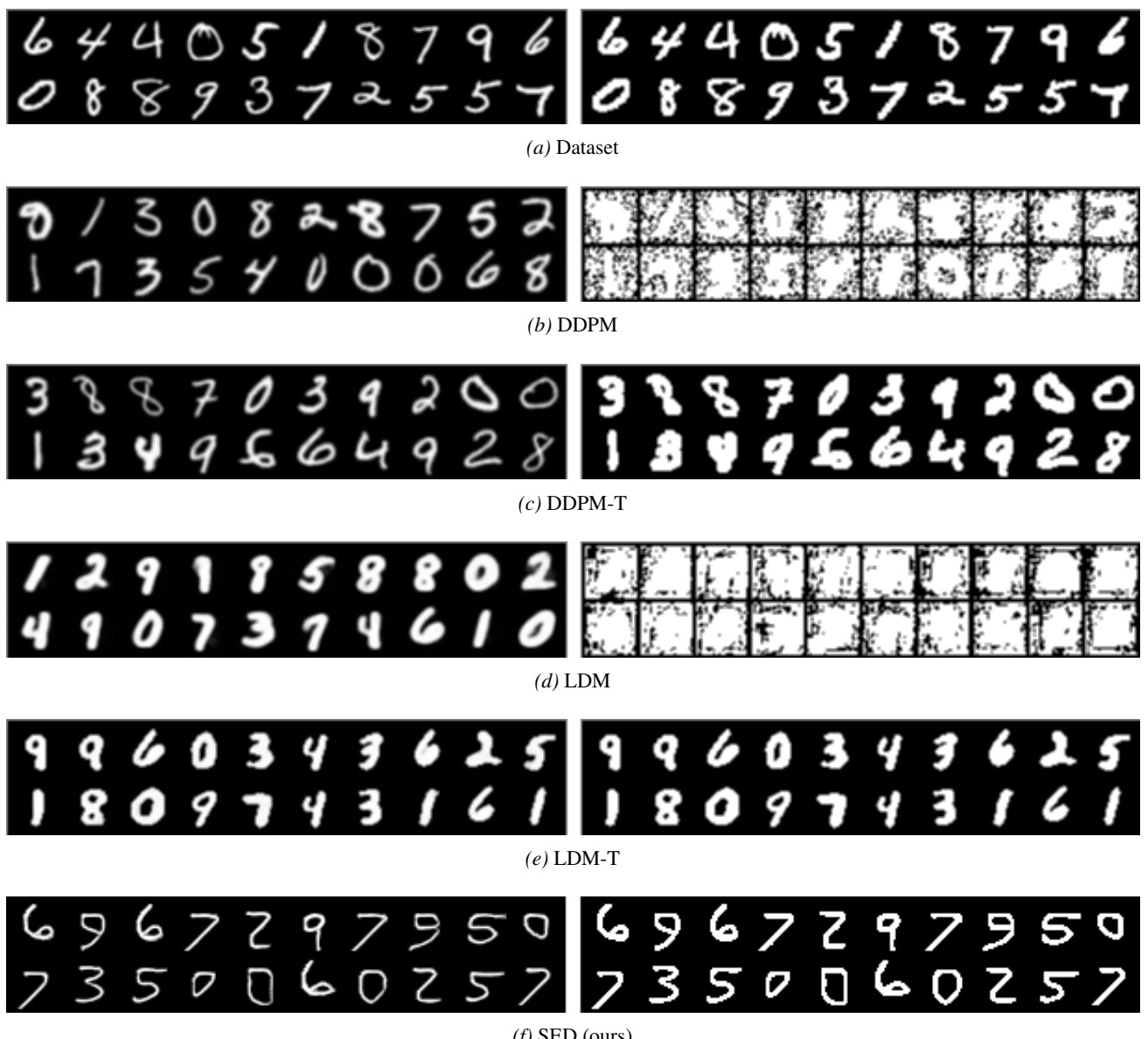

*(a)* Dataset

*(b)* DDPM

*(c)* DDPM-T

*(d)* LDM

*(e)* LDM-T

*(f)* SED (ours)

*Figure 12.* Shown are, from top to bottom, in the first row: MNIST images sampled from the dataset, DDPM sampled images, thresholded DDPM sampled images (DDPM-T), LDM sampled images, thresholded LDM sampled images (LDM-T), and SED sampled images. The second column contains the respective sparsity information. Despite highly visually similar images, DDPM and LDM fail to reflect the sparsity, whereas the proposed SED has a similar sparsity to the images from the dataset.

