# OpenReview forum: "Skipping the Zeros in Diffusion Models for Sparse Data Generation"
_ICML.cc/2026/Conference — ICML 2026 regular_

### Official Review · Reviewer_Crxt · 2026-03-04

**Soundness:** 4
**Presentation:** 3
**Significance:** 3
**Originality:** 3
**Overall Recommendation:** 5
**Confidence:** 4

**Summary:**

"Skipping the Zeros in Diffusion Models for Sparse Data Generation" presents a novel method for sparse data generation with a bespoke sparse latent diffusion model. The core idea is to avoid  explicitly modelling zeros altogether by operating on a compressed latent representation of only the sparse non-zero tokens, and decoding sparse (position, value) pairs autoregressively.
More precisely: sparse non-zero data is first extracted and encoded via a transformer encoder. This representation is mean-pooled, (analogous to pooled representations in transformer classifiers) to establish a full, single token representation. Within this space, latent diffusion is applied. After fully denoising, the denoised latent z from the LDM is passed through an autoregressive transformer decoder which generates (position, value) pairs until the [EOS] token is generated. Lastly, the sparse representation is converted back into the original representation (imaging or otherwise).

**Compliance With Llm Reviewing Policy:**

Affirmed.

**Final Justification:**

Following the authors’ rebuttal, I am updating my score from weak accept to accept.

The rebuttal was clear, thorough, and directly addressed my main questions. In particular, the clarification around mean pooling—both its role in mapping variable-length sparse inputs to a fixed-dimensional latent space and the empirical comparison with a [CLS] token—resolves my earlier concerns. While more expressive aggregation strategies (e.g., learned or attention-based pooling) were not extensively explored, the observation that [CLS] performs similarly suggests that additional complexity may not yield significant gains in this setting. Making this intuition explicit in the paper will further strengthen the design justification.

The authors also provided convincing evidence regarding autoregressive decoding across sequence lengths, showing that performance does not degrade with longer sequences and is instead more closely tied to data complexity. Additionally, the expanded discussion of related work (e.g., SLIDE and SDD) clarifies the paper’s positioning and highlights the distinct contribution of operating directly on compressed non-zero representations.

Overall, I find the proposed approach to be technically sound, conceptually clean, and practically meaningful. The idea of operating on compressed non-zero representations and performing diffusion in a fixed latent space is well-motivated and supported by empirical results. While there remain natural avenues for future work—such as learned pooling strategies, alternative generative formulations (e.g., flow matching or discrete diffusion), and potential end-to-end training—these do not detract from the core contribution.

In summary, this is a solid and well-executed paper that advances sparse data generation in a meaningful way, and I recommend acceptance.

**Key Questions For Authors:**

Could this be trained e2e? The paper implies that the SAVAE is kept fixed when training the latent diffusion model; while this is the primary choice for training LDMs, it would be more efficient and conceptually easier to implement if this could be trained end-to-end.

Why average the encoded latent? Wouldn't this dilute the input representation? Is it because you can have any number of sparse inputs? That point could be made a little clearer in the paper. This ties to the aforementioned pooling ablation.

How does the autoregressive decoding perform on sequences of different lengths? For example, one would expect a marginal reduction in quality for long sequences due to increasing the attention context length.

**Limitations:**

Yes

**Strengths And Weaknesses:**

Strengths

Encoding sparse representations directly and averaging the latent representation is a conceptually clean design choice that performs well compared to baseline methods.

Figure 4 does a lot of the architectural heavy lifting to understand the idea.

Figure 6 helps demonstrate improvements over other methods such as SARM and DDPM-Threshold which tend to predict much smaller signals, usually pixel-wise. This improvement is discussed line 323 right column.

A structural side effect of SED is that encoding and decoding scale compute directly with number of nonzero tokens. Additionally, latent diffusion operates in a fixed-dimensional space independent of the number of non-zero tokens. Therefore, this motivates the future consideration below.


Weaknesses / Comments

Concept of sparsely represented tokens for diffusion isn't by itself new [2], but the full pipeline seems sufficiently novel to recommend acceptance.

Comparison against [1] would be beneficial as it leverages both discretised and continuous inputs to help in the generation of sparse data. Also references to work from sparse 3D point clouds [2] would help position this paper against related works as these primarily differ in pre and post processing (plus minor differences like mean-pooling pre diffusion).

The sparse token averaging could do with an ablation to also briefly compare against using a cls token or an attention-informed weighted average or similar.





Citations:
[1] - Ostheimer, Phil, et al. "Sparse Data Diffusion for Scientific Simulations in Biology and Physics." EurIPS 2025 Workshop on SIMBIOCHEM.
[2] - Lyu, Zhaoyang, et al. "Controllable mesh generation through sparse latent point diffusion models." Proceedings of the IEEE/CVF conference on computer vision and pattern recognition. 2023.


Future considerations:


It may be worthwhile to analyse implicit methods or approaches that reduce the full 1000-step diffusion process for extremely sparse data, as the additional computation may not be necessary to adequately refine the latent representation z. For example, what performance is achieved by a single VAE objective (i.e., without diffusion) as a lower bound for this method?

---

> ### Author Rebuttal · Authors · 2026-03-30
>
> We thank the reviewer for the detailed and constructive feedback.
>
> ## Mean Pooling vs. [CLS] Token
>
> We evaluated both. As noted in line 185 (right column), using a [CLS] token yielded very similar performance to mean pooling. We chose mean pooling for simplicity and training stability.
>
> The reason pooling is needed at all: the number of non-zero entries varies across samples, so the encoder produces a variable number of token representations. Mean pooling maps this variable-length set to a single fixed-size vector, which is necessary for the subsequent fixed-dimensional diffusion process.
>
> We will clarify this reasoning in the camera-ready version.
>
> ## Autoregressive Decoding Across Sequence Lengths
>
> We do not observe systematic degradation with longer sequences. Table 4 provides evidence against this concern: FashionMNIST, which has the longest sequences (50% non-zero, \~392 tokens), achieves 100% correct dimension orderings. The highest error rates occur on the Muon datasets, which have shorter sequences (~50 tokens). This suggests that decoding difficulty depends on the complexity of the data distribution, not on sequence length alone. We will add this analysis to the paper.
>
> ## End-to-End Training
>
> We follow the standard two-stage latent diffusion paradigm [1,2]: train the autoencoder first, then train the diffusion model in the frozen latent space. This separation is the dominant approach in the LDM literature because it avoids coupling two optimization objectives, improving training stability. End-to-end training is possible in principle but would require balancing the reconstruction and diffusion losses simultaneously, and we leave this exploration to future work. We will clarify this choice in the paper.
>
> ## Comparison to Suggested Related Work
>
> **SLIDE [3]:** SLIDE represents 3D shapes as a fixed-size set of latent points. This is a form of structured, fixed-size sparsity. Our setting differs: the number of active dimensions varies per sample, and the data is a high-dimensional vector (not a 3D point cloud). We discuss related 3D point cloud work in lines 132-136 (especially [4]) and will expand the discussion to include SLIDE specifically.
>
> **SDD [5]:** SDD augments each dimension with a binary sparsity indicator and runs diffusion over the full augmented representation, doubling the input size. SED takes the opposite approach: it removes zero-valued dimensions before diffusion, so computation scales with the number of non-zeros. We will add this comparison to the related work section.
>
> We appreciate the reviewer’s positive assessment and believe these clarifications address the concerns raised. Therefore, we kindly ask the reviewer to reconsider the score in light of these points.
>
> [1] - Rombach, Robin, et al. "High-resolution image synthesis with latent diffusion models." Proceedings of the IEEE/CVF conference on computer vision and pattern recognition. 2022.
>
> [2] - Yang, Ling, et al. "Diffusion models: A comprehensive survey of methods and applications." ACM computing surveys 56.4 (2023): 1-39.
>
> [3] - Lyu, Zhaoyang, et al. "Controllable mesh generation through sparse latent point diffusion models." Proceedings of the IEEE/CVF conference on computer vision and pattern recognition. 2023.
>
> [4] - Luo, Shitong, and Wei Hu. "Diffusion probabilistic models for 3d point cloud generation." Proceedings of the IEEE/CVF conference on computer vision and pattern recognition. 2021.
>
> [5] - Ostheimer, Phil, et al. "Sparse Data Diffusion for Scientific Simulations in Biology and Physics." EurIPS 2025 Workshop on SIMBIOCHEM.

---

> > ### Author Rebuttal · Reviewer_Crxt · 2026-04-01
> >
> > Thank you to the authors for the clear and thorough rebuttal. I appreciate the detailed responses and the additional context provided.
> >
> > I would like to briefly clarify one of my earlier comments regarding the pooling choice. My question was not about why pooling is required (the explanation regarding variable-length sparse inputs mapping to a fixed-dimensional latent space is clear and reasonable), but rather why mean pooling is preferred over alternative aggregation strategies.
> >
> > In particular, I was curious whether more expressive approaches (e.g., attention-weighted pooling or learned aggregation) were considered, beyond the comparison to a [CLS] token. As the authors note, [CLS] performs similarly, which suggests that the diversity or complexity of the sparse representations may not be sufficient for different pooling mechanisms to yield significant differences. It would be helpful if this intuition were stated explicitly in the paper, as it strengthens the design justification.
> >
> > The rebuttal also satisfactorily addresses my questions regarding autoregressive decoding across sequence lengths and the positioning with respect to related work.
> >
> > However, despite this, and the other reviews; I’ve decided to increase my review from weak accept to accept. There are still areas that could be further refined and explored in future work, such as learned pooling strategies, flow matching approaches, and discrete diffusion methods, but these do not detract from the overall contribution of the paper.

---

> > > ### Author Response · Authors · 2026-04-08
> > >
> > > We thank the reviewer for the additional insights and for acknowledging the overall contributions of the paper. We will make the intuition about the pooling more explicit in the camera-ready version of the paper.

---

### Official Review · Reviewer_yY2G · 2026-03-11

**Soundness:** 3
**Presentation:** 2
**Significance:** 2
**Originality:** 2
**Overall Recommendation:** 4
**Confidence:** 3

**Summary:**

This paper studies generation of high-dimensional sparse continuous data where exact zeros are meaningful. The proposed SED pipeline uses a sparsity-aware VAE over non-zero dimension–value pairs, diffuses in dense latent space, and autoregressively decodes only non-zero entries. The motivation is clear from Figures 1–3 (pages 1–2), and the architecture in Figure 4 (page 4) is intuitive. Empirically, the method preserves sparsity much better than dense DDPM/LDM variants (Figure 5, page 6), improves physics metrics over baselines in Table 1 (page 6), and is competitive to strong biology baselines in Table 2 (page 7).

**Compliance With Llm Reviewing Policy:**

Affirmed.

**Key Questions For Authors:**

see above

**Limitations:**

yes

**Strengths And Weaknesses:**

Strengths.
1. The problem is real and underexplored. Many generative models do indeed treat zeros as just small continuous values, while in scientific data they often carry structural meaning.
2. it evaluates on calorimeter data, scRNA, and sparse vision, which makes the contribution feel broader than a narrowly engineered trick.
3. The paper is also generally well written.

Weaknesses.

1. My main reservation is that some efficiency and parameter-count claims feel stronger than the evidence supports. Table 3 (page 8) shows that SED is not uniformly faster: on Fashion-MNIST it is barely better or worse, so the abstract/conclusion should probably frame gains as high-sparsity-regime gains, not universal gains.

---

> ### Author Rebuttal · Authors · 2026-03-27
>
> We thank the reviewer for the positive assessment.
>
> ## Efficiency Gains and correction of Table 10
>
> We agree that efficiency gains are not universal and depend on the sparsity level. Table 3 confirms this for inferencing: on FashionMNIST (50% sparsity), SED sampling time (354.6 ms) is comparable to DDPM (357.7 ms) and slower than LDM (77.7 ms). In contrast, for highly sparse datasets (e.g., Muon Signal at 95%), SED is substantially faster (19× vs. DDPM and 4× vs. LDM). This dependency is already noted in lines 379–382 (right column), and we will revise the abstract and conclusion to clearly state that efficiency improvements primarily apply to high-dimensional, highly sparse settings.
>
> We also identified an error in Table 10 (efficiency gains for training) due to a transposed digit in the scRNA results (also **bold-faced**). The corrected values are shown below and further support the efficiency of SED in highly sparse regimes:
>
> |TrainingDataset|Sparsity|DDPM|LDM|SED|
> |---------------|--------|----|----|----|
> |Signal|95%|150.3|130.4|40.1|
> |Background|95%|151.0|131.3|40.1|
> |TabulaMuris|98%|10.2|10.1|**3.0**|
> |HumanLungPF|96%|10.4|10.3|**3.0**|
> |FashionMNIST|50%|123.8|161.5|280.1|
> |MNIST|81%|120.9|160.4|101.8|
>
>
> We hope that all questions are answered and encourage the reviewer to raise the score.

---

> > ### Author Rebuttal · Reviewer_yY2G · 2026-04-03
> >
> > resolved

---

### Official Review · Reviewer_3yXe · 2026-03-12

**Soundness:** 2
**Presentation:** 2
**Significance:** 2
**Originality:** 3
**Overall Recommendation:** 4
**Confidence:** 4

**Summary:**

The paper introduces Sparsity-Exploiting Diffusion (SED), a latent diffusion model tailored for generating high-dimensional, real-valued sparse data. Recognizing that standard diffusion models waste compute on zero-valued dimensions and fail to preserve exact sparsity patterns, the authors propose a two-stage approach. First, a Sparsity-Aware Variational Autoencoder (SAVAE) encodes only the non-zero dimension-value pairs into a compact, dense latent representation. Second, a standard diffusion process is trained on this dense latent space. During generation, the model denoises a sample from $\mathcal{N}(0,I)$ and uses an autoregressive Transformer decoder to reconstruct the sparse data by sequentially predicting dimension indices and their corresponding continuous values. The authors evaluate SED on particle physics calorimeter data, single-cell RNA (scRNA) sequencing data, and sparse variants of MNIST datasets.

**Compliance With Llm Reviewing Policy:**

Affirmed.

**Final Justification:**

My concerns have been resolved.

**Key Questions For Authors:**

How do you reconcile the core claim of achieving computational savings by "skipping zeros" with the empirical observation that SED takes approximately three times longer to train on highly sparse scRNA datasets compared to dense DDPM and LDM baselines?

Given that the autoregressive decoder imposes an artificial canonical ordering on non-sequential data, how does the model handle error accumulation during sampling?

Are the 12.1% of samples with invalid dimension orderings in the Muon Background dataset automatically discarded during your evaluation, or are they included in the Wasserstein distance calculations?

Could you justify the inconsistent parameter matching across domains (e.g., comparing a 15M parameter SED to a 37M parameter DDPM in physics, but a 4M parameter SED to a 5M parameter DDPM in biology)?

**Limitations:**

The authors explicitly acknowledge the limitation that SED generates incorrect dimension orderings during sampling, particularly noting the higher failure rates on the particle physics datasets.

**Strengths And Weaknesses:**

**Strengths**

The paper addresses a highly relevant and well-motivated problem. The observation that standard continuous diffusion models struggle to model exact zeros and subsequently erase domain-specific sparsity patterns is insightful and practically important for scientific disciplines.

The proposed architecture, particularly the concept of using a Transformer to extract non-zero values into a fixed-size dense latent space via a Dimension Encoding (DE), is a creative combination of zero-skipping techniques and latent diffusion.

**Weaknesses**

A central claim of the paper is that SED "delivers computational savings". However, the results on scRNA tabular data directly contradict this. The authors note that SED requires "approximately three times higher training time compared to DDPM and LDM" on datasets like Tabula Muris and Human Lung Pulmonary Fibrosis. This is attributed to the overhead of the Transformer architecture compared to efficient MLPs used by the baselines, fundamentally undermining the scalability claims for certain data modalities.

The SAVAE decoder relies on a canonical ordering of dimensions to autoregressively synthesize dimension-value pairs. This imposes an artificial sequential bias on unordered spatial or tabular data. Consequently, SED generates incorrect dimension orderings during sampling. For the Muon Background dataset, the model only achieves a correct ordering rate of 87.9%, meaning 12.1% of samples contain structural generation errors.

The parameter counts across models are highly inconsistent, complicating the evaluation of architectural benefits versus raw capacity. For physics and vision datasets, SED uses 15M parameters while the standard DMs use 37M parameters. Conversely, for the biology datasets, the SED U-Net uses 4M parameters compared to the 5M parameter baselines.

---

> ### Author Rebuttal · Authors · 2026-03-30
>
> We thank the reviewer for the thorough analysis. We address each concern below with concrete evidence.
>
> ## Computational Savings
>
> We agree that the term "computational savings" can be interpreted broadly. However, we believe the efficiency argument holds when the three important aspects of cost are examined separately:
>
> **1. Inference time (Table 3).** Diffusion models are typically trained once but sampled many times, making inference cost the more operationally relevant metric. SED is faster at inference on all highly sparse datasets.
>
> **2. Scaling with dimensionality (Figures 2, 9).** As input dimensionality grows from 1K to 27K genes, DDPM and LDM compute and memory scale linearly. SED remains nearly constant because it processes only non-zero entries. This is the core scaling advantage for high-dimensional sparse data.
>
> **3. Training time (Table 10).** We thank the reviewer for pointing this out. Upon re-checking, we identified a transposed digit in the reported scRNA training times (also **bold-faced**). After correction, SED is in fact ~3× faster, not slower, than the baselines on these datasets. The correct table is given below. We will fix this error in the camera-ready version.
>
> |TrainingDataset|Sparsity|DDPM|LDM|SED|
> |---------------|--------|----|----|----|
> |Signal|95%|150.3|130.4|40.1|
> |Background|95%|151.0|131.3|40.1|
> |TabulaMuris|98%|10.2|10.1|**3.0**|
> |HumanLungPF|96%|10.4|10.3|**3.0**|
> |FashionMNIST|50%|123.8|161.5|280.1|
> |MNIST|81%|120.9|160.4|101.8|
>
> In the camera-ready version, we will (1) qualify efficiency claims as applying to high-sparsity, high-dimensional regimes, (2) clearly separate training time from inference time and scaling behavior, and (3) revise Table 10.
>
> ## Dimension Ordering Errors and Their Impact
>
> **Are invalid samples filtered?** No. All reported metrics in Tables 1, 2, and Figures 5, 6 include samples with incorrect orderings. Despite the 12.1% ordering error rate on Muon Background, SED still achieves the best generation quality on that dataset ($W_P$ for $P_T$: 9.60 for SED vs. 12.61 for SARM vs. 21.77 for LDMI vs. 228.09 for DDPM). The ordering errors do not substantially degrade output quality.
>
> **Does error accumulation occur?** The evidence suggests ordering errors are driven by domain complexity, not sequence length. Muon Background has short sequences (\~5% non-zero, i.e., \~50 tokens for a 1024-dimensional input) but 12.1% errors. Tabula Muris has even higher sparsity (\~98%) but only 1.3-1.5% errors. FashionMNIST, which produces the longest sequences (\~50% non-zero), achieves 100% correct orderings. If errors accumulated with sequence length, we would expect the opposite pattern. The higher error rate on Muon datasets likely reflects the difficulty of modeling the combination of clustered and isolated energy deposits.
>
> ## Parameter Count Matching
>
> Parameter counts were matched to the corresponding latent diffusion baseline within each domain, which is the most controlled comparison since both SED and LDM are latent diffusion methods:
>
> - Physics/vision: SED = 15M, LDM = 15M (same budget). Input-space DDPM uses 37M because full-resolution CNNs require more capacity. SARM uses 25M.
> - Biology: SED = 4M, LDM = 4M (same budget). Input-space DDPM uses 5M.
>
> Under this controlled comparison, SED consistently outperforms LDM at equal parameter count across all datasets and metrics. Exact parameter matching across fundamentally different architectures (CNN vs. MLP vs. Transformer) is not feasible because the same parameter count yields different effective capacity in each architecture. We will state this design rationale explicitly in the revision.
>
> Overall, we hope these clarifications resolve the reviewer’s concerns, and we kindly ask the reviewer to reconsider the score in light of these explanations.

---

> > ### Author Rebuttal · Reviewer_3yXe · 2026-04-05
> >
> > My concerns have been resolved.

---

### Official Review · Reviewer_fAyc · 2026-03-13

**Soundness:** 3
**Presentation:** 3
**Significance:** 3
**Originality:** 3
**Overall Recommendation:** 4
**Confidence:** 4

**Summary:**

This paper proposes Sparsity-Exploiting Diffusion (SED), which models only the non-zero values in sparse data while preserving the sparsity structure. By skipping zero entries during both training and inference, SED reduces unnecessary computation and achieves computational savings while maintaining or improving generation quality. The approach is particularly suitable for domains with inherently sparse data, such as physics and biology.

**Compliance With Llm Reviewing Policy:**

Affirmed.

**Final Justification:**

The rebuttal address my main concerns.

**Key Questions For Authors:**

1. The paper adopts an autoregressive decoder in SAVAE to generate dimension–value pairs sequentially. However, for reconstruction-based objectives, it is more common to apply full attention over the latent features and reconstruct the output in parallel. The paper does not sufficiently justify why the autoregressive design is required here. Providing further explanation or ablation comparing autoregressive and non-autoregressive decoders would improve the clarity of this design choice.

2. The paper evaluates SAVAE mainly through sample-level sparsity preservation, but does not report any metrics for the reconstruction performance of the autoencoder itself. Could the authors provide quantitative evaluation of the reconstruction quality (e.g., reconstruction error, likelihood, or RMSE) to better assess how well SAVAE preserves the original sparse data?

**Limitations:**

See weakness

**Strengths And Weaknesses:**

**Strengths:**

1.  The paper identifies a clear mismatch between diffusion models and sparse data, namely that dense architectures waste computation on zero-valued dimensions. This is an important and underexplored issue for scientific datasets.

2. The core idea—explicitly skipping zero entries and modeling only non-zero dimension-value pairs—is conceptually simple yet effective. It aligns with classical sparse representations (e.g., CSR/CSC formats).

**Weakness:**

1. The title appears somewhat misleading. It implies that zero values are skipped directly during the diffusion denoising process. In contrast, the proposed method primarily removes zero entries through a sparsity-aware VAE that extracts non-zero dimension–value pairs, after which diffusion operates on the resulting latent representation. Therefore, the “skipping” occurs mainly at the representation level rather than within the diffusion process itself. The authors may consider revising the title or clarifying this point in the introduction.

2. The design of the decoder in SAVAE is not clearly described (see questions below). Besides, the paper only reports sample-level sparsity preservation, while no metrics are provided to assess the reconstruction quality of the autoencoder itself. Including reconstruction metrics (e.g., reconstruction error or likelihood) would help better understand the effectiveness of SAVAE.

---

> ### Author Rebuttal · Authors · 2026-03-30
>
> We thank the reviewer for the constructive feedback.
>
> ## Title
>
> We thank the reviewer for pointing this out. The current title may suggest that zeros are skipped inside the diffusion process, whereas sparsity handling occurs at the representation level. We will remove "Skipping the Zeros" from the title in the camera-ready version.
>
> ## Why the SAVAE Decoder Must Be Autoregressive
>
> The autoregressive design is a structural necessity, not a modeling preference. The reason is that each sample has a different number of non-zero entries: for example, one scRNA cell may have 50 active genes, while another may have 200. At generation time, the model does not know in advance how many (dimension, value) pairs to produce. It must generate them one by one until it emits an [EOS] token.
>
> A parallel (non-autoregressive) decoder would require fixing the output length to a single value for all samples, which contradicts the variable-sparsity setting. Alternatively, it would require a separate length-prediction module, adding complexity with no clear benefit.
>
> Importantly, the autoregressive design does not slow down training for highly sparse datasets. We use teacher forcing (lines 695-697), which allows parallel computation over all tokens in a sequence. Sequential generation is only needed at sampling time.
>
> We will add this justification to the paper.
>
> ## Reconstruction Quality of SAVAE
>
> We provide MSE reconstruction error comparing SAVAE against a standard VAE:
>
> **Train MSE Loss**
> | Dataset | VAE | SAVAE |
> |---|---:|---:|
> | Muon Signal | 3.3 $\times 10^{-6}$ | 2.5 $\times 10^{-6}$ |
> | Muon Background | 4.5 $\times 10^{-6}$ | 4.1 $\times 10^{-6}$ |
> | Tabula Muris | 1.8 $\times 10^{-4}$ | 0.4 $\times 10^{-4}$ |
> | Human Lung PF | 1.2 $\times 10^{-4}$ | 0.6 $\times 10^{-4}$ |
> | FashionMNIST | 7.0 $\times 10^{-3}$ | 8.1 $\times 10^{-3}$ |
> | MNIST | 5.1 $\times 10^{-3}$ | 3.4 $\times 10^{-3}$ |
>
> **Validation MSE Loss**
> | Dataset | VAE | SAVAE |
> |---|---:|---:|
> | Muon Signal | 3.5 $\times 10^{-5}$ | 3.3 $\times 10^{-5}$ |
> | Muon Background | 7.2 $\times 10^{-6}$ | 6.5 $\times 10^{-6}$ |
> | Tabula Muris | 1.9 $\times 10^{-4}$ | 0.4 $\times 10^{-4}$ |
> | Human Lung PF | 1.3 $\times 10^{-4}$ | 0.6 $\times 10^{-4}$ |
> | FashionMNIST | 7.3 $\times 10^{-3}$ | 8.4 $\times 10^{-3}$ |
> | MNIST | 3.2 $\times 10^{-3}$ | 3.75 $\times 10^{-3}$ |
>
> SAVAE achieves lower reconstruction error than a standard VAE on all highly sparse datasets (sparsity > 80%). The only exception is FashionMNIST (50% sparse), where the longer output sequences reduce the advantage of the sparse representation. We will include this table in the camera-ready version.
>
> We hope that all concerns are covered and encourage the reviewer to raise the score.

---

> > ### Author Rebuttal · Reviewer_fAyc · 2026-04-01
> >
> > My concerns have been adequately addressed.

---

### Decision · Program_Chairs · 2026-04-30

**Decision:**

Accept (regular)

**Comment:**

Reviewers universally appreciated the problem of handling sparsity in
diffusion models.  They found it to be a well-executed study of the
problem, leading to significant inference time savings in sparse
datasets.  A good paper for the conference.